# Occurrence of *Escherichia coli* Pathotypes in Diarrheic Calves in a Low-Income Setting

**DOI:** 10.3390/pathogens12010042

**Published:** 2022-12-27

**Authors:** Wagaw Sendeku Chekole, Haileeyesus Adamu, Susanna Sternberg-Lewrein, Ulf Magnusson, Tesfaye Sisay Tessema

**Affiliations:** 1Department of Clinical Sciences, Swedish University of Agricultural Sciences (SLU), 75007 Uppsala, Sweden; 2Institute of Biotechnology, Addis Ababa University, Addis Ababa 1176, Ethiopia; 3Institute of Biotechnology, University of Gondar, Gondar 196, Ethiopia; 4Department of Biomedical Sciences & Veterinary Public Health, Swedish University of Agricultural Sciences, 75007 Uppsala, Sweden

**Keywords:** diarrhea, calves, PCR, *E. coli*, pathotype

## Abstract

Different *E. coli* pathotypes are common zoonotic agents. Some of these pathotypes cause recurrent and widespread calf diarrhea and contribute to significant economic losses in the livestock sector worldwide in addition to putting humans at risk. Here, we investigated the occurrence of *E. coli* pathotypes in diarrheic calves in Ethiopia kept under various calf management practices. One hundred fecal samples were collected from diarrheic calves in 98 different farms. *E. coli* was isolated in the samples from 99 of the diarrheic calves, and virulence genes were detected in 80% of the samples. The occurrence of *E. coli* pathotypes in the samples was 32% ETEC, 23% STEC, 18% STEC/ETEC, 3% EPEC, 2% EAEC, and 1% EHEC. No diarrheic calves were positive for the EIEC and DAEC pathotypes. The occurrence of pathotypes was positively associated with female calves (EPEC, *p =* 0.006), aged less than 2 weeks (STEC, *p =* 0.059), and calves fed colostrum via the hand method (STEC, *p =* 0.008 and EAEC, *p =* 0.003). This study revealed that several *E. coli* pathotypes occurred among calves affected with diarrhea. Moreover, the presence of a mixed STEC/ETEC pathotypes infection was present in the studied low-income setting. These findings indicate a considerable risk for the zoonotic transmission from calves to humans and the options to provide the better management for younger calves in order to reduce the economic loss.

## 1. Introduction

*Escherichia coli* is widely regarded as a commensal bacterium [1]. Nonetheless, some *E. coli* pathotypes are pathogens causing intestinal infections ranging from mild to severe in animals and humans [2,3]. Common pathotypes include Enterohaemorrhagic *E. coli* (EHEC), Enterotoxigenic *E. coli* (ETEC), Enteropathogenic *E. coli* (EPEC), Enteroaggregative *E. coli* (EAEC), Enteroinvasive *E. coli* (EIEC), and diffusely adherent *E. coli* (DAEC) [4]. Among these pathotypes, the EHEC serotype O157:H7 is recognized as the most important zoonotic pathogen. Globally, between 2007 to 2015, EPEC- and ETEC-related yearly human deaths amounted to 37,000 and 26,000, respectively [5]. The World Health Organization (WHO) has estimated that one-third of the population in low-income countries suffers from foodborne diseases, of which pathogenic *E. coli* is one cause [5].

In low- and middle-income countries, calf diarrhea caused by *E. coli* is common and causes a considerable number of deaths [6,7,8]. For instance, diarrheal diseases caused up to 15% of preweaning calf mortality in Uruguay [9] and 31% [10] and 63% [11] in Ethiopia, where calf diarrhea is an issue, though it is also an issue in high-income countries. For instance, in the USA, where about 57% of weaning calf mortalities have been reported due to diarrhea, with the majority of the cases in calves less than one month old [12]. About 9% mortality was reported in male calves with diarrhea in Canada [13]. In Norway, the overall calf mortalities were 280,000 heads per year, and the economic loss was estimated to be USD 10 million in 2006 [14]. Overall, calf diarrheal disease mainly seemed to be associated with poor farming practices [15,16]. 

Thus, an infection with *E. coli* pathotypes in calves poses a significant risk to animal health with a considerable impact on productivity and the farmers’ economy, in addition to being a public health issue due to their zoonotic potential. Given the generally poorer hygienic conditions and disease prevention capacity in rural areas in low-income countries, these risks may be accentuated in such settings. However, data on the occurrence of different *E. coli* pathotypes in low-income countries is sparse [17,18,19,20]. Here, we provide such data from diarrheic calves in a district in Ethiopia, the most livestock-rich country in Africa [21].

## 2. Materials and Methods 

### 2.1. Study Area and Period 

The study was conducted in Basona Werana woreda (District), Ethiopia (Figure 1). The Basona District is found in the north Shewa zone of the Amhara regional state, 130 Km from the capital, Addis Ababa. The district is divided into 22 small administrative subdistricts (Kebeles). The district has a population of 120,930 individuals living in 27,686 households. The households are distributed in urban and rural areas with an urban to a rural proportion of 0.01 [22]. It has an annual temperature range of 6 to 20 °C and the rainfall varies between 950 and 1200 mm. The farming practice is a mixed crop–livestock (crops, cattle, sheep, and goat) type where cattle predominate.

### 2.2. Study Design and Sampling Technique 

A cross-sectional study design was used. The sample collection was carried out from October 2020 to March 2021, following the contact of veterinarians in each subdistrict. Farm owners were informed about the sampling through their milk union and requested to report the occurrence of diarrhea in calves. All types of farms with diarrheic calves were included. Each farm was visited once during the study period. Samples were taken from diarrheic calves between 0 and 10 weeks old. The age groups were 0–2, 2–4, 4–7, and 7–10 weeks. One fecal sample was collected from one calf per farm, except on one occasion when three diarrheic calves were sampled on a single farm. The sample size was set based on the financial constraints and a reasonable representation of the farms.

A total of 100 fecal samples from diarrheic calves were collected from 98 farms. Five ml feces were collected from the rectum of the calves. All samples were temporarily stored in phosphate-buffered saline (HiMEDIA, Mumbai, India) and transported to Addis Ababa University, the Institute of Biotechnology, health biotechnology laboratory, and processed within 24 h of the collection time. 

### 2.3. Isolation and Characterization of E. coli 

The isolation and characterization of *E. coli* were performed using standard bacteriological procedures [23]. Briefly, all of the collected samples were enriched in tryptose soya broth (TSB) (HiMedia, Mumbai, India) at 37 °C for 18–24 h. Subsequently, a loopful of this bacterial culture was streaked onto McConkey agar (HiMedia, Mumbai, India) and was grown at 37 °C for 18–24 h. The pink color colonies (lactose fermenter) from the McConkey agar culture were picked and transferred onto an Eosin Methylene Blue (EMB) (HiMedia, Mumbai, India) agar medium and incubated to grow at 37 °C for 18–24 h. The *E. coli* colonies were presumptively identified based on their appearance of a greenish metallic sheen on Eosin Methylene Blue (EMB) (HiMedia, Mumbai, India) medium. Three well-isolated presumptive *E. coli* colonies were randomly selected from each EMB plate and pure cultures from the selected isolates were used for further testing. The *E. coli* isolates were identified using the biochemical tests Indole, Methyl Red, Vogues–Proskauer test, and Citrate utilization (IMViC) [23]. 

### 2.4. DNA Extraction 

The DNA was extracted from the *E. coli* isolates as described by Moore et al. [24] with a minor modification. Briefly, a single colony was subcultured in Luria Bertani (LB) broth media (HiMEDIA, Mumbai, India). A 1.5 mL bacterial culture was pelleted via centrifugation and subsequently resuspended and washed in 100 µL sterile deionized water. The 150 µL bacterial suspensions were lysed at 100 °C and then frozen at −20 °C. Finally, the suspensions were centrifuged at 10,000 rpm for 10 min, and 100 µL of supernatant containing DNA from each preparation was transferred into fresh Eppendorf tubes. The DNA preparations were quantified using a Nano-drop spectrophotometer and a ratio of 260 nm/280 nm was used to estimate the quality of the DNA. Moreover, gel electrophoresis was used to check the intactness of the isolated DNA.

### 2.5. E. coli Pathotyping Using PCR

The DNA from the *E. coli* isolates was subjected to a PCR amplification of ten virulence genes; bundle-forming Pili (*bfp*), attaching and effacing (*eae*), adhesion (*daaA-E*); shiga-toxin-1/2 (*stx-1/2*), heat-stable (*st*) and heat-labile (*lt*) toxins, hemolysin toxin (*hly*), virulence protein transporter (*aatA),* and invasion-associated protein (*ial*) using the primers listed in Table 1 and the *E. coli* isolates were classified into one of the six pathotypes based on their virulence genes. 

All primers and PCR conditions used for the virulence gene amplification were performed as described previously [25,26,27,28,29,30,31,32,33]. All of the PCR reactions were performed in a singleplex platform in a Prima 96 plus Thermal Cycler (Himedia Laboratories, Mumbai, India). 

Each PCR reaction was carried out in a 25 μL final volume reaction mixture containing nuclease-free water (Himedia, Mumbai, India), 10X PCR buffer with 17.5 mM of MgCl_2_ (Himedia, Mumbai, India), 0.35 mM of each dNTPs (Himedia, Mumbai, India), 100 pM for each of the virulence gene-specific forward and reverse primers (Bioneer; Daejeon, Republic of Korea), 500 U Taq polymerase enzyme (DELTA Biotechnology, Ethiopia), and a DNA template. DNA samples that carried the relevant virulence gene(s) and nuclease-free water (without template) were used as the positive and negative control, respectively. PCR amplification products were processed using 1.5% agarose gel electrophoresis at 100 Volt for one hour; the gels were stained with ethidium bromide in 1× TAE buffer (40 mM Tris-HCl, 20 mM Acetate, 0.5 mM EDTA, pH 8.3). A DNA ladder of 100 bp (Himedia, Mumbai, India) was run in parallel with the PCR products to determine the size of the amplicons. Separated PCR products were visualized, photographed, and documented in a gel documentation system (Bio-Rad, Feldkirchen, Germany).

### 2.6. Farm Data

A structured questionnaire was prepared and used to collect the relevant farm data (Appendix A). In the questionnaire, farm owners were given closed-ended questions about the overall management of the calves. The questions covered the sex, age, and breed of the calf, colostrum and supplementary feeding practice, housing type, and flooring. In addition to the questionnaire, data about the hygiene was collected through a direct observational judgment by the first author. To grade the hygiene of the housing, a four-point scale ranging from 1 to 4, 1—very poor, 2—poor, 3—good, and 4—very good, was used. For the housing hygiene evaluation, a checklist was used to minimize the judgment biases, containing the following aspects: proper manure disposal, dryness, aeration, and the absence of nearby waste, each with 1 point (Appendix A). The questionnaire and observational data as well as the fecal samples from diarrheic calves were collected during the same visit to the farm. Each farm was visited once between October 2020 and March 2021. 

### 2.7. Data Processing and Analysis 

Data related to the sex, age, breed, colostrum feeding status, time of the colostrum feeding, type of supplementary feed, housing type, and hygiene were coded and entered into a Microsoft^®^ Excel spreadsheet. The responses to the multiple-choice questions were coded into categorical variables. Similarly, the “Yes” or “No” responses were categorized as “1” or “0”, respectively. The housing hygiene “Very poor” to “Very good” responses were coded into numbers ranging from 1 to 4. 

Descriptive statistics were used to analyze the occurrence of virulence genes and *E. coli* pathotypes. The questionnaire responses and their association with the proportion of detected *E. coli* pathotypes were analyzed using the chi-square test, with *p <* 0.05 considered as a significant association. A pairwise analysis of the co-occurrences of virulence genes was carried out using a two-tailed chi-square test, with *p <* 0.05 seen as a strong correlation. The software SPSS (IBM SPSS Statistics 28.0.0.0) was used for the analyses. 

## 3. Results

### 3.1. Farm Description and Calf Management 

A total of 100 diarrheic calves from 98 farms of three different types, family (95), enterprise (2), and research (1) were sampled. The size of the farms ranged from 2 to 32 animals with an average of 6.5 and a median value of 6 animals. The farms were grouped into small farms (SF, *n* = 51) with 2–6 animals and medium farms (MF, *n* = 47) with 7–32 animals (Appendix A). 

Of the sampled calves, 53% were male and of all of the calves, 3% were less than 1 week old, 30% were between 2 and 4 weeks old, 27% were between 4 and 7 weeks old, and 39% were between 7 and 10 weeks old. Crossbreds (local X Holstein Friesians) were the most common, accounting for 85% of the calves. About 89% of the calves were fed colostrum, and 43% of these were fed colostrum within 6 h after their birth. Allowing calves to suckle colostrum was the most common practice, with this occurring in 72% of farms. Twenty-seven (23%) of the calves were also provided additional feeds by grazing and hay, respectively. In 31% of the farms, calves were housed in the same barn with other animal species. The majority of the houses (74%) had soil flooring and 63% of the calving houses were judged to have poor hygienic conditions (Appendix A). 

### 3.2. E. coli Isolation, Virulence Genes (VGs) Detection, and E. coli Pathotyping

The results of the *E. coli* isolation, PCR pathotyping, and infection multiplicity are shown in Figure 2. Of all the 300 presumptive *E. coli* isolates from 100 diarrheic calves, 281 were confirmed as *E. col.* Among the isolated *E. coli*, 160 were identified as an *E. coli* pathotype. *E. coli* was isolated from 99 sampled calves and, of these, 79 were found to be infected with at least one of the *E. coli* pathotypes. Of the infected calves, 61 were infected with single-pathotype and 18 had mixed *E. coli* pathotypes.

The gel electrophoresis demonstrating the PCR products of virulence genes are shown in Figure 3. The samples from 99 diarrheic calves were assessed for the presence of virulence gene/s (Table 2). The proportions of detected virulence genes were; *eae* 3 (3%), *stx1* 8 (8.1%), *stx2* 9 (9%), *hly* 1 (1%), *aatA* 2 (2%), and *st* 30 (30.3%). Several virulence genes were found in combinations; the most common combinations were *stx2-st*, 10.1% (*n* = 10), *sx1-st, 6.1*% (*n* = 6), *stx1-stx2*, 6.1% (*n* = 6), and *lt-st*, 2% (*n* = 2). Similarly, *stx1-stx2*-st and *stx2-hly* were each found in 1% (*n* = 1) of the diarrheic calves. The *bfp*, *ial*, and *daaE* virulence genes were not found in any of the tested samples. No virulence genes were found in the samples from 20.2% (*n* = 20) of the diarrheic calves. 

Of the 79 diarrheic calves positive for *E. coli* pathotypes, ETEC 32.3% (*n* = 32), STEC 23.3% (*n* = 23), and STEC/ETEC 18.2% (*n* = 18) were the most prevalent pathotypes. On the other hand, EHEC, EAEC, and EPEC pathotypes were less common in diarrheic calves with 1, 2, and 3% detection rates, respectively. EPEC pathotypes were atypical, lacking the *bfp* virulence gene. In this study, no diarrheic calves were positive for the EIEC and DAEC pathotypes. 

Table 3 shows the distribution of the *E. coli* pathotypes based on the different farm sizes, sex, age, and breed of the diarrheic calves. The number of detected *E. coli* pathotypes was higher on medium-size farms (MF), in female calves, calves under 2 weeks of age, and crossbred calves. In diarrheic calves from MF, 4.1% (2) EPEC and 20.8% (10) STEC/ETEC were found, the corresponding figures from small farms (SF) were EPEC 2.0% (1) and 15.7% (6) STEC/ETEC (15.7%). All of the detected EPEC pathotypes were from female calves (*p =* 0.059). Calves under 2 weeks of age had the highest incidence of STEC, 100%, (3/3), while the 2–4 weeks-old calves had the lowest incidence 13.3%, (4) (*p =* 0.006). Calves of an indigenous breed were more commonly infected with mixed STEC/ETEC, 33.3% (*n* = 5) than crossbreed calves 15.5% (*n* = 13) pathotypes.

The detected *E. coli* pathotypes among isolates from calves fed colostrum and a different supplementary feed are presented in Table 4. Among calves that had not been fed colostrum, the STEC, (27.3%, *n* = 3) and STEC/ETEC (36.4%, *n* = 4) pathotypes were more common; the corresponding results from the calves which were fed colostrum were 22.7% (*n* = 20) and 15.9% (*n* = 14), respectively. The detection rates of most *E. coli* pathotypes were higher in calves fed colostrum > 6 h from birth as compared to calves fed colostrum within 6 h after birth. STEC were detected in 8 of 17 (47.1%) of calves who were fed colostrum by hand, which was higher than in the 12 of 71 (16.9%) of calves who were allowed to suckle colostrum themselves (*p =* 0.008). The EAEC pathotypes were found in 2 of 17 calves who were fed colostrum by the hand method (*p =* 0.003). *E. coli* pathotypes were more frequently detected from calves that had not been started on a supplementary feed and calves who were allowed to graze than from calves who were fed concentrate and hay, but the observed differences were not significant. 

The detection rate variations between *E. coli* pathotypes in calves from different housing types, flooring, and hygienic status were minor and no significant associations were seen (Table 5). *E. coli* pathotypes were present in 25 of 30 (83.3%) of the diarrheic calves housed in barns with other animals. Calves in houses with a soil flooring had higher *E. coli* pathotype proportions (80.6%) than calves on concrete (77.8%) and stone-lined (77.8%) floors. The *E. coli* pathotypes were found in 100%, 80%, 76.5%, and 73.3% of calves housed in very good, poor, very poor, and good hygienic conditions, respectively.

The distribution of pathotypes between the study subdistricts is shown in Figure 4. ETEC pathotypes were detected in 9 of the 10 subdistricts studied, with about 46.9% (*n* = 15) from Angolela and 53.1% (*n* = 17) of the pathotypes from eight other subdistricts. Compared to other subdistricts, STE/ETEC, 37.5% (*n* = 3), appeared more frequently among the isolates from the Debre Birhan subdistrict. EHEC (*n* = 1) was only found from the Wushawushign subdistrict (*p =* 0.00).

The results from the pairwise correlation analysis between the detected virulence genes are shown in Figure 5. The correlation analysis revealed that the co-occurrence of virulence genes *stx2-lt, stx1-stx2, stx2-hly*, and *lt-st* virulence genes had a weak positive correlation. The majority of virulence genes showed weak negative correlations. The observed correlations were not statistically significant.

## 4. Discussion 

The present study intended to describe the occurrence of *E. coli* pathotypes and overall calving practices in a low-income setting. Livestock production is important in many Sub-Saharan countries like Ethiopia, for the national economy as well as for the livelihood of individual families, but the sector is facing various challenges. This study supports previous reports that diarrhea in calves is one of these challenges [15].

Virulence genes are extensively used in determining *E. coli* pathotypes and have been found in multiple combinations [34,35]. In our study, *stx1,* stx2, and *st* VGs were more frequent than *eae*, *hly*, *aatA*, and *lt*. In another report, however, *stx1* and *eae* were found at a higher rate than *st* and *stx2* [35]. In the isolates where virulence genes were found in combination, *stx2-st*, *sx1-st, stx1-stx2*, and *lt-st* represented the dominant combinations. In addition, *stx1-stx2*-st and *stx2-hly* were identified in a low proportion of samples from diarrheic calves. None of the samples were positive for the *bfp*, *ial*, and *daaE* virulence genes, in contrast to a previous report that indicated the presence of these genes in *E. coli* from diarrheic calves [36]. 

When translating the VRG combinations into pathotypes, ETEC was detected from 32.3% of the diarrheic calves as a sole pathogenic isolate. This result was in accordance with Dall et al. [37] who described ETEC as the leading cause of severe neonatal calf diarrhea in Brazil. All of the ETEC pathotypes carried the *st* virulence gene. It has been reported that *st* encodes structurally stable and highly virulent toxins [38]. Other studies also suggested that ETEC carries multiple chromosomal and plasmid virulence gene combinations that enhance the fitness and transmission between hosts [39,40]. Moreover, a study in Egypt described an *st*-associated ETEC pathotype in diarrheic calves [41,42]. The present finding and the previous studies indicate that ETEC plays a major role in causing calf diarrhea. However, the results presented in this study contradicted that of Umpiérrez et al. [34] and Khawaskar et al. [43] who reported only 10% and 1.9% ETEC from diarrheic calves in Uruguay and India.

In the present study, STEC was the second most frequent pathotype found in 23.2% of diarrheic calves. This is in agreement with previous reports that STEC is a key pathotype in calf diarrhea [2,43]. In addition, STEC has been detected at a higher rate in diarrheic calves [40] and other diarrheic animals than in healthy animals [44]. Several studies [44,45,46] have reported that STEC isolates with *stx1* were more common than those that carried *stx2*, which contradicts the findings in the current study. Similar with the current study, Sobhy et al. [47] showed that the majority of STEC isolates had *stx2* instead of *stx1* and *stx1-stx2* combinations. These inconsistent results indicate that *stx1, stx2*, and *stx1-stx2* play a role in causing diarrhea among calves. In addition, STEC was reported in 34% of non-diarrheic water buffalo calves in Brazil [48] and 5.5% of healthy cattle calves in Korea [35]. This is in line with previous suggestions that animals are important reservoirs of STEC [35,49]. Moreover, the severity of the disease could be associated with the number and type of VGs present, suggesting that an infection by *E. coli* isolates with more VGs in various possible combinations are more likely to cause a disease. 

This study revealed that 18.2% of the sampled calves had mixed STEC and ETEC (STEC/ETEC) infections. This result was in line with Awad et al. [41] who reported ETEC/STEC (14.7%) and ETEC/EPEC (2.7%) in diarrheic calves in Egypt. Other researchers showed that diarrheic calves could be infected with a hybrid STEC-ETEC pathotype [50,51]. These pathogroups are characterized as hetero-pathotypes comprised of virulence genes representing two or more *E. coli* pathotypes [52]. Other studies reported that various hetero-pathotypes, EAEC-STEC and EPEC-STEC, successfully colonized newborn calves and caused persistent diarrheal diseases [53,54]. 

In the present study, EPEC, EAEC, and EHEC pathotypes were found only in 3, 2, and 1% of diarrheic calves, respectively. These low percentages are in line with the findings from diarrheic neonatal calves sampled in India: EPEC (2.9%), EAEC (0.9%), and EHEC (0.3%) [43]. A study of diarrheic water buffalo diarrheic calves in Brazil detected EPEC (3.4%) and EHEC (3.4%) [48]. However, the 2% EAEC detection rate in our study is lower than the 39% EAEC detection rate in the same study [48]. This difference could be due to differences in the study’s design or target population (the species, health status, and geographical location). 

In the current study, we attempted to relate the occurrence of pathotypes in the fecal samples to some basic farm and calf management practices. However, these observations must be interpreted with caution as the sampling focused on clinical cases with no healthy controls and was not randomized, which would have been needed for a solid epidemiological analysis. The incidence of *E. coli* pathotypes was higher in diarrheic calves sampled from medium sized farms. Regardless of the farm’s size, calves ≤ 2 weeks old were more frequently infected than other age groups. This is similar to previous studies that noted that younger calves were more frequently infected with different *E. coli* pathogroups than older calves [55,56]. This might be due to differences between younger and older calves regarding their immunity and rumen function. In addition, this could be due to age-related changes in the receptor expression for adhesins of the different pathotypes in the intestinal epithelial cells. In the present study, more female and crossbreed diarrheic calves were found to be infected with *E. coli* pathotypes. There is no plausible explanation for these differences, and the sex skewed finding is inconsistent with a report from India where the detection of STEC (8.0%) and ETEC (2.5%) were higher in samples from male neonatal calves. This was suggested due to the fact that male calves did not receive appropriate care as they were considered economically less important [43]. 

Our results show that *E. coli* pathotypes were more common in calves not fed colostrum than in calves fed colostrum. We also noted that calves fed colostrum 6–24 h after birth were more frequently infected by *E. coli* pathotypes than those fed within 0–6 h. Multiple studies show that the sufficient and timely colostrum feeding of calves improves the growth, immunity, and intestinal microbiota of the calves [57,58], as maternal antibodies in the colostrum prevent a bacterial attachment to the intestinal epithelium [59,60,61]. According to the findings in this study and multiple other studies, the colostrum feeding of calves should takes place within 6 h after their birth. Of the 17 diarrheic calves fed colostrum via hand, 47.1% (*p* = 0.008) and 11.8% (*p =* 0.003) were positive for STEC and EAEC, respectively. Similar research in Ethiopia has indicated that hand/bucket feeding was significantly correlated with a high incidence of *E. coli* [15]. The higher occurrence of pathotypes in samples from hand-fed calves compared with those that suckled colostrum may indicate a lack of appropriate hygiene routines during the hand-feeding of the colostrum to calves. This merits further study.

The detection of *E. coli* pathotypes was slightly higher in calves housed in barns (with other animals) and on soil floors. This result was in agreement with Fesseha et al. [15]. We also found that calves housed with very good hygiene had a higher frequency of *E. coli* pathotypes; however, those farms were comparatively larger farms, where a higher infectious pressure with *E. coli* pathotypes could be assumed. The interaction between the farm’s size and hygiene may have prevented the full exploration of these aspects in our limited number of samples. 

## 5. Conclusions

The study showed that *E. coli* pathotypes could be frequently found in diarrheic calves in a low-income setting. The ETEC and STEC pathotypes were the majority of pathotypes found among the diarrheic calves. A significant number of calves were found to be infected with mixed STEC/ETEC pathotypes. Given the rearing conditions in these settings, where calves are kept close to human dwellings, this poses a significant zoonotic risk. 

The presence of *E. coli* pathotypes was higher in calves under 2 weeks of age and crossbred calves. Calves that had not been fed colostrum and were late-fed colostrum appeared to be more vulnerable to *E. coli* pathotypes. This, together with the frequent occurrence of *E. coli* pathotypes, suggests there is a need for providing the better management for calves and younger calves in particular to improve their health and reduce the economic loss. 

## Figures and Tables

**Figure 1 pathogens-12-00042-f001:**
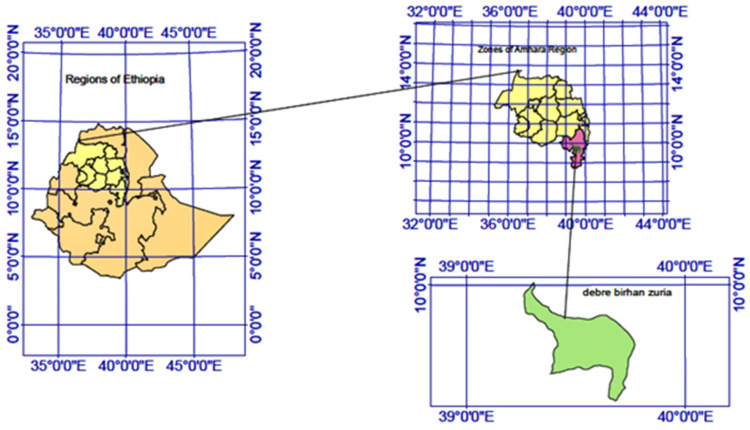
Study area map. The map on the left-hand side (yellow) shows Ethiopia; the upper right shows Amhara regional state and the lower r right in green shows the study area known as Basona Werena district (woreda).

**Figure 2 pathogens-12-00042-f002:**
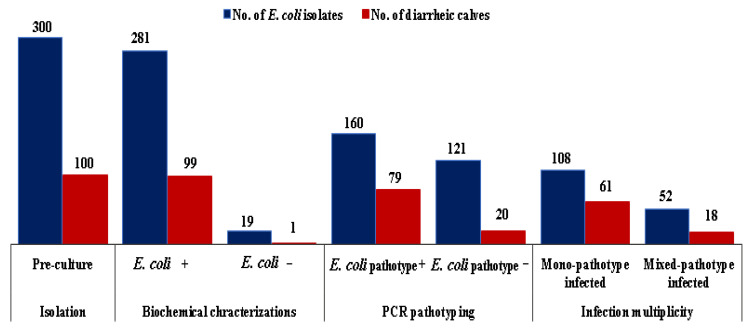
Number of *E. coli* isolates and number of sampled calves after initial isolation, confirmation, and pathotyping and detected infection multiplicity.

**Figure 3 pathogens-12-00042-f003:**
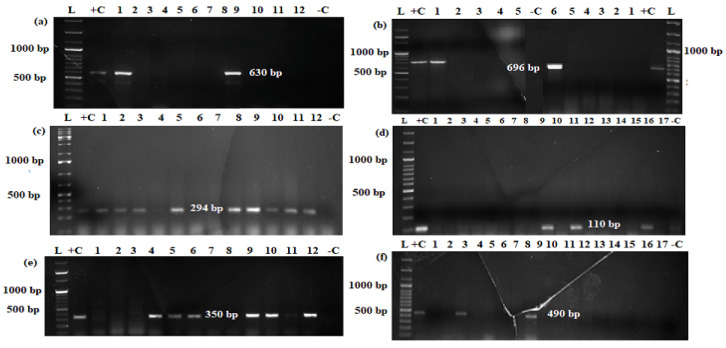
Representative gel electrophoresis results of PCR products for virulence genes: (**a**) *aatA* (630 bp), (**b**) *lt* (696 bp), (**c**) *st* (294 bp), (**d**) *stx1* (110 bp), (**e**) *stx2* (350 bp), and (**f**) *eae* (490 bp). L-ladder DNA (marker size, 100 bp plus), Lane +C-positive control, lanes with numbers are PCR products of virulence genes from *E. coli* isolates (test) and lane −C-negative control.

**Figure 4 pathogens-12-00042-f004:**
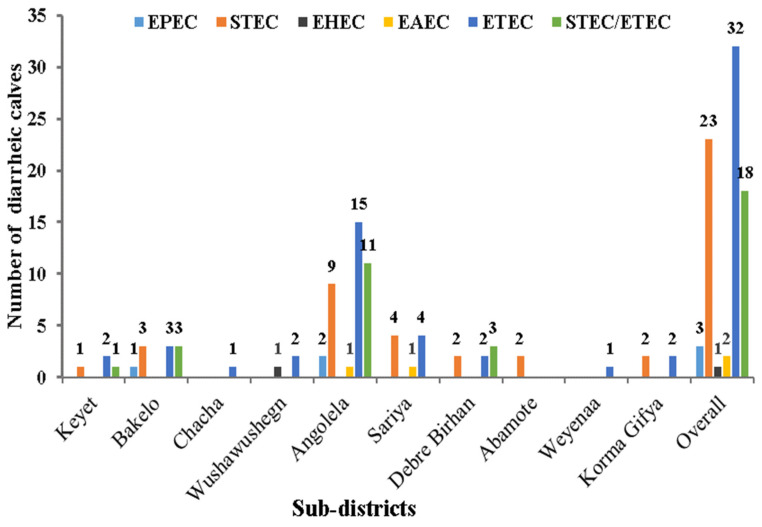
Distributions of *E. coli* pathotypes in 10 study subdistricts from fecal samples of 99 diarrheic calves in Basona Werana district, Ethiopia. Pathotypes; EPEC—Enteropathogenic *E. coli*; STEC—Shiga-toxin *E. coli*; EAEC—Enteroaggregative *E. coli*; ETEC—Enterotoxigenic *E. coli*; and STEC/ETEC–STEC/ETEC (mixed Shiga and stable toxins) *E. coli*.

**Figure 5 pathogens-12-00042-f005:**
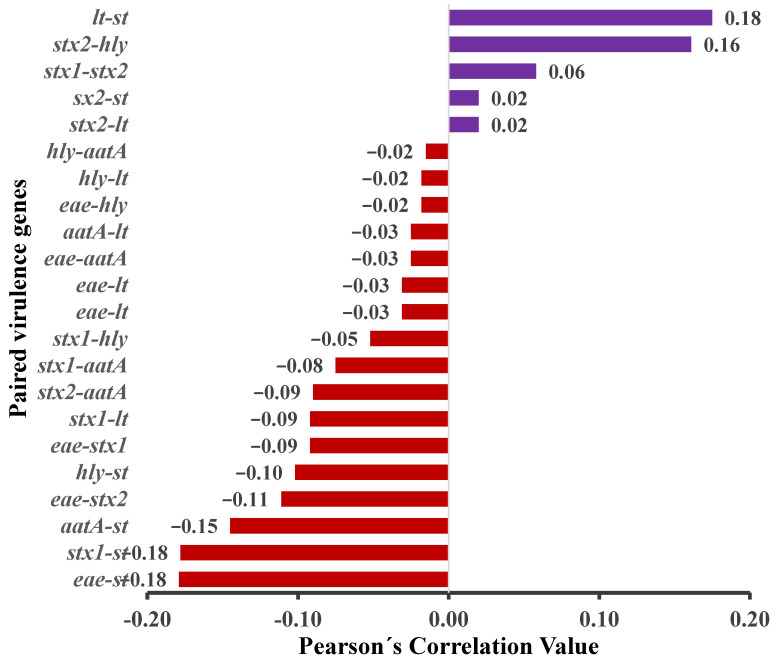
The pairwise correlation coefficient of detected *E. coli* virulence genes in the 99 diarrheic calve’ fecal isolates in Basona Werana district Ethiopia. *eae*-intimin; *stx ½*-Shiga toxin 1&2; *hly*-hemolysin; *aatA*-aggregative, *st*-stable toxin; *lt*-liable toxin.

**Table 1 pathogens-12-00042-t001:** *Escherichia coli* pathotyping oligonucleotide primers: primer sequences for the ten (10) virulence genes and amplicon size for each of the six E. coli pathotypes are shown below.

Primer	Oligonucleotide Sequence 5′ to 3′	Gene	Pathotype	Product Size (bp)	Reference
EAE1 EAE2	F:AAACAGGTGAAACTGTTGCC R:CTCTGCAGATTAACCTCTGC	*eae*	EPEC/EHEC	490	[25]
EVS1 EVC2	F:ATCAGTCGTCACTCACTGGT R:CTGCTGTCACAGTGACAAA	*stx1*	STEC/EHEC	110	[26]
EVT1 EVT2	F:CAACACTGGATGATCTCAGC R:CCCCCTCAACTGCTAATA	*stx2*	STEC/EHEC	350	
EHEC EHEC	F:ACGATGTGGTTTATTCTGGA R:CTTCACGTCACCATACATAT	*hly*	EHEC	167	[27]
EAEC EAEC	F: CTGGCGAAAGACTGTATCAT R:CAATGTATAGAAATCCGCTGTT	*aatA*	EAEC	630	[28]
BFP BFP	F:AATGGTGCTTGCGCTTGCTGC R:GCCGCTTTATCCAACCTGGTA	*bfpA*	Typical EPEC	324	[29]
ST1 ST2	F: TTT ATT TCT GTA TTG TCT T R:GCAGGATTACAACACAATTC	*St*	ETEC	294	[30]
LT1 LT2	F: GGCGACAGATTATACCGTGC R: CCGAATTCTGTTATATATGTC	*lt*	ETEC	696	[31]
IAL F IAL R	F: CTGGATGGTATGGTGAGG R:GGAGGCCAACAACATTATTTCC	*ial*	EIEC	320	[32]
daaE 1 daaE 2	F:GAACGTTGGTTAATGTGGGGT R:TATTCACCGGTCGGTTATCAG	*daaE*	DAEC	542	[33]

F—forward primer, R—reverse primer, and bp—basepair.

**Table 2 pathogens-12-00042-t002:** Virulence genes profiles and distribution of *E. coli* pathotypes in fecal samples from 99 diarrheic calves on 98 farms in Basona Werana district, Ethiopia.

Pathotype	Virulence Genes							Diarrheic Calves	Total
	*eae*	*stx1*	*stx2*	*hly*	*aatA*	*lt*	*st*	*n* (%)	*n* (%)
EPEC	+		-	-	-	-	-	3 (3.0)	3 (3.0)
STEC	-	+	-	-	-	-	-	8 (8.1)	23 (23.2)
	-	+	+	-	-	-	-	6 (6.1)	
	-	-	+	-	-	-	-	9 (9.1)	
EHEC	-	-	+	+	-	-	-	1 (1.0)	1 (1.0)
EAEC	-	-	-	-	+	-	-	2 (2.0)	2 (2.0)
ETEC	-	-	-	-	-	+	+	2 (2.0)	32 (32.3)
	-	-	-	-	-	-	+	30 (30.3)	
STEC/ETEC	-	+	-	-	-	-	+	6 (6.1)	18 (18.2)
	-	-	+	-	-	-	+	10 (10.1)	
	-	+	+	-	-	-	+	1 (1.0)	
	-	-	+	-		+	+	1 (1.0)	
TPD	-	-	-	-	-	-	-	-	79 (79.8)
NPD	-	-	-	-	-	-	-	-	20 (20.2)
Overall	-	-	-	-	-	-	-	-	99 (100.0)

*n* &%—number and percentage of calves positive for tested virulence genes and *E. coli* pathotypes; TPD—total pathotypes detected; NPD—no pathotypes detected; virulence genes; *eae*- intimin; *stx ½*—Shiga toxin 1&2; *hly*—hemolysin; *aatA*—aggregative; *st*—stable toxin; *lt*—liable toxin); pathotypes; EPEC—Enteropathogenic *E. coli*; STEC—Shiga-toxin *E. coli*; EAEC—Enteroaggregative *E. coli*; ETEC—Enterotoxigenic *E. coli* and STEC/ETEC—mixed STEC and ETEC *E. coli* pathotype.

**Table 3 pathogens-12-00042-t003:** Detection of *E. coli* pathotypes from different farm sizes, sex, age, and breed of 99 diarrheic calve fecal isolates in Basona Werana district, Ethiopia.

Pathotype					Diarrheic Calves—*n* (%)					
	Farm Size		Sex		Age (Weeks)				Breed	
	SF	MF	F	M	≤2	(2–4]	(4–7]	(7–10]	Indigenous	Crossbreed
EPEC	1 (2.0)	2 (4.2)	0 (0.0 ^b^_)_	3 (6.5 ^b^)	0 (0.0)	0 (0.0)	1 (3.7)	2 (5.1)	0 (0.0)	3 (3.6)
STEC	13 (25.5)	10 (20.8)	13 (24.5)	10 (21.7)	3 (100.0 ^a^)	4 (13.3 ^a^)	5 (18.5 ^a^)	11 (28.2 ^a^)	3 (20.0)	20 (23.8)
EHEC	1 (2.0)	0 (0.0)	0 (0.0)	1 (2.2)	0 (0.0)	1 (3.3)	0 (0.0)	0 (0.0)	0 (0.0)	1 (1.2)
EAEC	1 (2.0)	1 (2.1)	1 (1.9)	1 (2.2)	0 (0.0)	0 (0.0)	1 (3.7)	1 (2.6)	0 (0.0)	2 (2.4)
ETEC	16 (31.4)	16 (33.3)	16 (30.2)	16 (34.8)	0 (0.0)	12 (40.0)	8 (29.6)	12 (30.8)	4 (26.7)	28 (33.3)
STEC/ETEC	8 (15.7)	10 (20.8)	11 (20.8)	7 (15.2)	0 (0.0)	7 (23.3)	5 (18.5)	6 (15.4)	5 (33.3)	13 (15.5)
NPD=	11 (21.6)	9 (18.8)	12 (22.6)	8 (17.4)	0 (0.0)	6 (20.0)	7 (25.9)	7 (17.9)	3 (20.0)	17 (20.2)
Total (*n*)	51	48	53	46	3	30	27	39	15	84

row values with the same letter superscript ^a^ and ^b^ were significantly different; at *p =* 0.006 and 0.059, respectively. SF—Small farm (≤6 animals/farm); MF—medium farm (≥7 animals/farm); F—female; M—male; T (*n*)—total number of calves examined in the categories; *n* (%)—number and percentage of calves positive for respective *E. coli* pathotypes. NPD—no pathotypes detected; EPEC—Enteropathogenic *E. coli*; STEC—Shiga-toxin *E. coli*; EAEC—Enteroaggregative *E. coli*; ETEC—Enterotoxigenic *E. coli* and STEC/ETEC—mixed STEC and ETEC pathotypes.

**Table 4 pathogens-12-00042-t004:** Distribution of *E. coli* pathotypes from diarrheic calves in relation with colostrum feeding status (*n* = 99), colostrum feeding time and method (*n* = 88), and supplementary feed (*n* = 99) in Basona Werana district of Ethiopia.

Pathotype	Diarrheic Calves—*n* (%)										
	Colostrum Feeding						Supplementary Feed				
	Feeding		Feeding Time (Hours)		Feeding Method						
	No	Yes	(0–6]	(6–24]	Suckle	Hand	NS	Gz	Con	Hay	Com
EPEC	0 (0.0)	3 (3.4)	1 (2.3)	2 (4.4)	3 (4.2)	0 (0.0)	0 (0.0)	2 (7.7)	0 (0.0)	0 (0.0)	1 (4.0)
STEC	3 (27.3)	20 (22.7)	8 (18.6)	12 (26.7)	12 (16.9 ^b^)	8 (47.1 ^b^)	3 (18.8)	9 (34.6)	0 (0.0)	5 (21.7)	6 (24.0)
EHEC	0 (0.0)	1 (1.1)	0 (0.0)	1 (2.2)	1 (1.4)	0 (0.0)	0 (0.0)	0 (0.0)	0 (0.0)	0 (0.0)	1 (4.0)
EAEC	0 (0.0)	2 (2.3)	0 (0.0)	2 (4.4)	0 (0.0)	2 (11.8 ^a^)	0 (0.0)	1 (4.0)	0 (0.0)	0 (0.0)	1 (4.0)
ETEC	2 (18.2)	30 (34.1)	16 (37.2)	14 (31.1)	27 (38.0)	3 (17.6)	7 (43.8)	5 (19.2)	4 (44.4)	9 (39.1)	7 (28.0)
STEC/ETEC	4 (36.4)	14 (15.9)	5 (11.6)	9 (20.0)	11 (15.5)	3 (17.6)	4 (25.0)	4 (15.4)	2 (22.2)	4 (17.4)	4 (16.0)
NPD	2 (18.2)	18 (20.5)	13 (30.2)	5 (11.1)	17 (23.9)	1 (5.9)	2 (12.5)	5 (19.2)	3 (33.3)	5 (21.7)	5 (20.0)
Total (*n*)	11	88	43	45	71	17	16	26	9	23	25

row values with the same letter superscript ^a^ and ^b^ were significantly different at *p =* 0.003 and 0.008, respectively. NS—not started; GZ—grazing: Con—concentrates; Com—combined; T (*n*)—total number of calves examined in the categories; *n* (%)—number and percentage of calves positive for respective *E. coli* pathotypes; NPD—no pathotypes detected; EPEC—Enteropathogenic *E. coli*; STEC—Shiga-toxin *E. coli*; EAEC—Enteroaggregative *E. coli*; ETEC—Enterotoxigenic *E. coli* and STEC-ETEC—mixed STEC and ETEC pathotypes.

**Table 5 pathogens-12-00042-t005:** The distribution of *E. coli* pathotypes from diarrheic calves kept in different house types and hygienic practices in Basona Werana district, Ethiopia. The variations in the distribution of *E. coli* pathotypes across housing situations were minor and no significant difference values were seen.

Pathotype	Diarrheic Calves—*n* (%)																	
	Housing Type				Housing Floor						Housing Hygiene							
	Pen		Barn		Soil		Concrete		Other		Very Poor		Poor		Good		Very Good	
	*n*	%	*n*	%	*n*	%	*n*	%	*n*	%	*n*	%	*n*	%	*n*	%	*n*	%
EPEC	1	1.4	2	6.7	2	2.8	1	11.1	0	0.0		0.0	2	3.3	0	0.0	1	14.3
STEC	15	21.7	8	26.7	15	20.8	3	33.3	5	27.8	5	29.4	13	21.7	2	13.3	3	42.9
EHEC	1	1.4	0	0.0	1	1.4	0	0.0	0	0.0	0	0.0	1	1.7	0	0.0	0	0.0
EAEC	2	2.9	0	0.0	1	1.4	0	0.0	1	5.6	0	0.0	2	3.3	0	0.0	0	0.0
ETEC	24	34.8	8	26.7	25	34.7	1	11.1	6	33.3	2	11.8	22	36.7	7	46.7	1	14.3
STEC/ETEC	11	15.9	7	23.3	14	19.4	2	22.2	2	11.1	6	35.3	8	13.3	2	13.3	2	28.6
NPD	15	21.7	5	16.7	14	19.4	2	22.2	4	44.4	4	23.5	12	20.0	4	26.7	0	0.0
Total (*n*)	69	100.0	30	100.0	72	100.0	9	100.0	18	100.0	17	100.0	60	100.0	15	100.0	7	100.0

T (*n*)—total number of calves examined in the categories; *n*—number and %—the percentage of calves positive for respective *E. coli* pathotypes; NPD—no pathotypes detected; EPEC- Enteropathogenic *E. coli;* STEC- Shiga-toxin *E. coli*; EAEC—Enteroaggregative *E. coli;* ETEC—Enterotoxigenic *E. coli* and STEC/ETEC—mixed STEC and ETEC pathotypes.

## Data Availability

The data generated will be provided upon the request of the corresponding author.

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
