# Peer review of "Occurrence of Escherichia coli Pathotypes in Diarrheic Calves in a Low-Income Setting"

_pathogens, 2022, doi:10.3390/pathogens12010042_

Round 1
Reviewer 1 Report
I had the pleasure of reading this, and found it an interesting and well reported document. Pathogenic E. coli is a growing issue in farming, and can become a big problem across the industry, so further information on this is always welcome. The zoonotic potential is also important here, and is well explained in the study.
I have a few comments below.
Line 16- perhaps instead of besides, maybe as well as, or in addition to may sound better?
Line 44- likely better as calf mortality rather than calve
Line 129- this is often referred to as a no template control
Line 174- It would be nice to see more included here on the age split of the animals, and how that affects infection rates?
Line 185- I am surprised at the lack of testing of healthy animals on the same farms. What is the chances that some of the animals may be asymptomatically infected? Perhaps could have been worth including
Check E. coli is in italics in the legends of most figures- figure 2, Table 2, table 3, Figure 4, Figure 5
Line 194- can delete ‘of’- several virulence genes …..
Table 3- is it possible to highlight the significant results in the table for ease of understanding?
Line 261- different housing types may sound better?
Line 262-263- E. coli needs to be in italics
Line 281- genes is a typo (gens)
Line 308- calf may sound better than calve?
Line 333- were these animals more significantly ill or died compared to single infections?
Line 347- What would be the reason for this difference?
Line 361- maybe read better as- This was suggested to be due to the fact that male calves …..
Line 366- you mention timely delivery of colostrum- is there a specific best time to deliver it?
Line 373- it would be interesting to follow this up and assess the E. coli on hands, buckets etc as a transmission vehicle, and work out the best way to disinfect them
Line 384- were the majority among…. (reword)
Line 391- coli needs a small c.
Author Response
Response to Reviewer 1
Comments and Suggestions for Authors
I had the pleasure of reading this and found it an interesting and well-reported document. Pathogenic E. coli is a growing issue in farming and can become a big problem across the industry, so further information on this is always welcome. The zoonotic potential is also important here and is well explained in the study.
I have a few comments below.
Comment: Line 16- perhaps instead of besides, maybe as well as, or in addition to may sound better?
Response: Thanks, this has been corrected accordingly. Line 16
Comment: Line 44- likely better as calf mortality rather than calve
Response: Thank you, this has been corrected with minor modifications. Line 46 - 47
Comment: Line 129- this is often referred to as a no-template control
Response: We thank you for the concept. The concept is explained accordingly. Line 135
Comment: Line 174- It would be nice to see more included here on the age split of the animals, and how that affects infection rates?
Response: Thank you for the comment, the information is provided accordingly. Lines 180 - 182
Comment: Line 185- I am surprised at the lack of testing of healthy animals on the same farms. What is the chance that some of the animals may be asymptomatically infected? Perhaps could have been worth including
Response: The authors acknowledge the concern. The study was designed to determine the occurrence of E. coli pathotypes only in calves with diarrheal cases and hence did not include the sampling of healthy calves. We fully acknowledge that clinically healthy animals could be infected asymptomatically. Perhaps this aspect could be addressed in the future. Lines 193 - 194
Comment: Check E. coli is in italics in the legends of most figures- figure 2, Table 2, table 3, Figure 4, Figure 5
Response: We apologise for this oversight; the issue is corrected throughout the manuscript.
Comment: Line 194- can delete ‘of’- several virulence genes …..
Response: Thanks, corrected. Line 206
Comment: Table 3- is it possible to highlight the significant results in the table for ease of understanding?
Response: We thank the reviewer. Significant results are highlighted in Tables 3 and 4.
Comment: Line 261- different housing types may sound better?
Response: Thank you for the comment. it is corrected with minor modifications. Lines 274 - 276
Comment: Line 262-263- E. coli needs to be in italics
Response: Sorry, this has been corrected. Lines 265 - 266
Comment: Line 281- genes is a typo (gens)
Response: Apologies, corrected. Line 300
Comment: Line 308- calf may sound better than calve?
Response: Thank you for the comment. It has been corrected accordingly. Line 327
Comment: Line 333- were these animals more significantly ill or died compared to single infections?
Response: We did not trace back how mixed pathotype-infected calves were ill or how the rate of mortality varied compared to single pathotype infected calves. it is an interesting and important question that would have required sampling more calves on each farm and is an important question that needs to be addressed. However, we are unable to provide the relevant data now. Lines 352 - 353
Comment: Line 347- What would be the reason for this difference?
Response: We appreciate the comment. This could be due to geographical variations or differences between the host species studied but is most likely due to differences in study design. These reasons are, however, speculations. A comment has been added. Lines 366 - 369
Comment: Line 361- maybe read better as This was suggested to be due to the fact that male calves …..
Response: Thanks, done accordingly. Line 386
Comment: Line 366- you mention timely delivery of colostrum- is there a specific best time to deliver it?
Response: Thank you for the comment. Based on our finding the best time of colostrum feeding to the calves was within 6 hours from birth. Sampled calves were allocated into two groups of colostrum feeding time: (0 – 6] and (6 – 24] hours after birth. E. coli pathotypes were more frequently detected from calves fed colostrum after 6 hours of birth compared to within 6 hours of birth. Therefore, according to the finding of this study and as suggested by multiple other studies the timely colostrum feeding to the calves is within 6 hours after birth. Lines 394- 396
Comment: Line 373- it would be interesting to follow this up and assess the E. coli on hands, buckets etc as a transmission vehicle, and work out the best way to disinfect them
Response: Thank you for the comment. Indeed, we agree. However, such a study needs to be repeated with different study designs (relatively larger sample size, more geographical locations with different colostrum feeding practices). Lines 396 - 402
Comment: Line 384- were the majority among…. (reword)
Response: We appreciate the comment, corrected. Line 412
Comment: Line 391- coli needs a small c.
Response: We apologize, and done accordingly. Line 419
Reviewer 2 Report
Overall, this is a clear, concise, and well-written manuscript. The introduction is relevant and theory based. Sufficient information about the previous study findings is presented for readers to follow the present study. The content is technically sound, and overall, the research is well described. The conclusions are supported by the analysis of the data presented. I only have one suggestion to delete on line 263 “and no…” (is repeated).
I really enjoyed reading this article, and I don’t have any special issues to comment on.
Author Response
Response to Reviewer 2
Comments and Suggestions for Authors
Overall, this is a clear, concise, and well-written manuscript. The introduction is relevant and theory based. Sufficient information about the previous study findings is presented for readers to follow the present study. The content is technically sound, and overall, the research is well described. The conclusions are supported by the analysis of the data presented.
Comment: I only have one suggestion to delete on line 263 “and no…” (is repeated).
Response: Sorry, this mistake, has now been adjusted accordingly. Line 279
Comment: I really enjoyed reading this article, and I don’t have any special issues to comment on.
Response: It sounds great to us, thank you for the comments.
Reviewer 3 Report
Review, paper no. pathogens-2085345 entitle „Occurrence of Escherichia coli pathotypes in diarrheic calves in a low-income setting”. This is a well-organized study, with sufficient methodology and adequate description of the results. Authors' research has shown interesting relationships. The manuscript idea is somehow new with great interest and it is well written as well. The authors have used the standard journal format in manuscript writing. The manuscript contains several inaccuracies in methodology. The advantage is that a large number of farms have been tested.
Specific comments:
Abstract:
Is sufficiently presented (methods, results, general conclusions).
Introduction: The introduction section is sufficient and analytically and adequately covers the need for the study.
Methods: The methodology is sufficiently presented. However, it has a few inaccuracies.
Line 79. Accurately determine the age of the calves. The first 3 months of life are the most interesting. How many samples have been taken during this period.
Results: The results of the study are analytically presented. Figures are adequate explain the findings of the study.
Provide in the results specific information regarding the timing and amount of colostrum given.
Improve the quality of figures.
Discussion: The results of study are sufficiently discussed.
Line 365. Revise for clarity
Could authors define possible limitations of the study? These are just analytical limits.
Conclusion: In conclusion, generalizations are given.
Author Response
Response to Reviewer 3
Comments and Suggestions for Authors
Review, paper no. pathogens-2085345 entitle „Occurrence of Escherichia coli pathotypes in diarrheic calves in a low-income setting”. This is a well-organized study, with sufficient methodology and adequate description of the results. Authors' research has shown interesting relationships. The manuscript idea is somehow new with great interest and it is well written as well. The authors have used the standard journal format in manuscript writing. The manuscript contains several inaccuracies in methodology. The advantage is that a large number of farms have been tested.
Specific comments:
Methods: The methodology is sufficiently presented. However, it has a few inaccuracies. Line 79. Accurately determine the age of the calves. The first 3 months of life are the most interesting. How many samples have been taken during this period.
Response: Faecal samples were taken from 100 diarrheic calves aged between 0 to 10 weeks across 98 farms. The sampled calves were grouped into 4 age groups: (0 – 2], (2 – 4], (4 – 7], and (7 – 10] weeks old. This information has been added. Lines 83 - 85.
Results: The results of the study are analytically presented. Figures are adequate explain the findings of the study. Provide in the results specific information regarding the timing and amount of colostrum given.
Response: Two groups of colostrum feeding times within 6 hrs from birth and 6 – 24 hrs were studied. the results related E. coli pathotypes concerning the two colostrum feeding times are shown in Table 4. The amount of colostrum given to the calves was noted and we are not able to provide this information at the moment.
Comment: Improve the quality of figures.
Response: we thank you for the comment. figure qualities are improved.
Comment: Line 365. Revise for clarity
Response: Thanks, done accordingly. Line 388
Comment: Could authors define possible limitations of the study? These are just analytical limits.
Response: Thank you for the comment. The limitations we have mentioned focus on study design/sampling and are the main limitations. We are not sure if more have to be included in the text.
Comment: Conclusion: In conclusion, generalizations are given.
Response: We acknowledge this comment and have rephrased the first and last sentences in the Conclusions slightly to address this. Lines 411, 419
Round 2
Reviewer 1 Report
I wish to thank the authors for their comments hard work.
I think that the manuscript reads well, and I am happy to recommend publication